# Perceptions of Change after a Trauma and Perceived Posttraumatic Growth: A Prospective Examination

**DOI:** 10.3390/bs9010010

**Published:** 2019-01-15

**Authors:** Adriel Boals, Lee A. Bedford, Jennifer L. Callahan

**Affiliations:** Department of Psychology, University of North Texas, Box 311280, Denton, TX 76203, USA; leebedford@yahoo.com (L.A.B.); jennifer.callahan@unt.edu (J.L.C.)

**Keywords:** posttraumatic growth, distress, coping, trauma

## Abstract

Recent research has distinguished between actual posttraumatic growth (PTG) and perceived PTG. We used a prospective research design to measure both actual and perceived PTG in an attempt to replicate and extend previous findings. We examined college students (*N* = 64) who experienced a traumatic event between the start (Time 1) and end (Time 2) of a semester. We included three measures of change from pre- to post-trauma: (1) Actual PTG (change scores in measures of PTG domains), (2) perceived general growth (Time 2 ratings of functioning at Time 1 subtracted from actual ratings given at Time 1), and (3) perceived PTG (self-reports of PTG on the posttraumatic growth inventory). The results revealed perceived general growth and actual PTG were significantly correlated, suggesting that participants’ perceptions of change were accurate. However, perceived PTG was not significantly related to either actual PTG or perceived general growth. Further, increases in actual PTG and perceived general growth were significantly related to decreases in distress and unrelated to coping. By contrast, higher levels of perceived PTG were significantly related to increases in distress and higher levels of avoidance coping. Our results suggest perceived PTG may be more of a coping process than an accurate recall of posttraumatic change.

## 1. Introduction

Since its introduction in the mid-1990s, a plethora of research has focused on the possibility that people may experience positive changes after a traumatic event (Tedeschi and Calhoun, 1996, [1]). Such positive changes have been referred to by a number of names, including benefit finding and adversarial growth, but the most commonly used name is posttraumatic growth (PTG). The concept of PTG theorizes that some people experience marked improvements as a direct result of experiencing a traumatic event in domains such as personal strength, appreciation of life, relationships with others, new possibilities, and spiritual growth (Tedeschi and Calhoun, 1996 [1]). The popularity of PTG has soared in the past decade, resulting in a plethora of research studies and its incorporation into clinical practices (Roepke, 2015; Shakespeare-Finch, and Lurie-Beck, 2014; Wu et al., 2019, [2,3,4]).

Research in psychology gives us reason to suspect that self-reports of PTG may be biased and sometimes even illusory (Boals and Schuler, 2018; Jayawickreme and Blackie, 2014; Frazier et al., 2009, [5,6,7]). For instance, research in autobiographical memory has found that people generally perceive themselves as improving over time (Brown, Buckner, and Hirst, 2011; Wilson and Ross, 2001, [8,9]). If an individual experiences a stressful, traumatic, or life-altering experience, is it easy for that individual to misattribute their perceived growth over time as resulting from said event (Coyne and Tennen, 2010, [10]). Additionally, empirical evidence finds that people’s perception of how their personality has changed over time is, at best, only modestly related with any actual personality change (Costa and McCrae, 1989; Ward and Wilson, 2015; Wilson and Ross, 2001, [9,11,12]) and is often skewed towards beliefs of positive, self-enhancing changes (Fleeson and Heckhausen, 1997, [13]). Another reason to suspect self-reports of PTG to be biased is that research in social psychology is full of cautionary tales demonstrating that humans have a strong tendency to overinflate their perceptions of themselves (Greenwald, 1981; Humberg et al., 2018; Taylor and Brown, 1988, [14,15,16]), and this tendency is exacerbated in times of distress (McFarland and Alvaro, 2000, [17]). Cognitive biases that contribute to such overinflated views of the self include dissonance reduction, motivated reasoning, positive illusions, self-serving attributions, and self-enhancement (Gilbert, Pinel, Wilson, Blumberg, and Wheatley, 1998, [18]). Social psychologist Daniel Gilbert calls this collection of cognitive biases the ‘psychological immune system’ (Gilbert et al., 1998, [18]). The psychological immune system is designed to maintain a sense of well-being and ego during times of adversity and distress. It is very possible that such biases in views of the self and biased perceptions of changes in the self over time affect self-reports of PTG. The cumulative result is that self-reports of PTG are vulnerable to biases and inaccuracies, leading to possible illusory perceptions of how much respondents believe they have grown as a result of a traumatic event.

The possibility of biased reporting has led to a growing movement that questions the validity of self-reports of PTG (Jayawickreme and Blackie, 2014, [6]). A meta-analysis examined the relationships between self-reports of PTG and various mental health measures we would expect to have strong relationships with PTG, including depression, anxiety, global distress, and quality of life (Helgeson, Reynolds, and Tomich, 2006, [19]). The results found that self-reports of PTG were either weakly related or unrelated to these mental health measures. This lack of criterion validity has fueled questions of construct validity. The vast majority of PTG studies use the posttraumatic growth inventory (PTGI; Tedeschi and Calhoun, 1996, [1]) to assess levels of PTG. The PTGI is a self-report measure of PTG that asks respondents to report the extent to which they believe they have grown as a result of a traumatic event. In a prospective study of the construct validity of the PTGI, Frazier et al. (2009), [7], administered assessments of functioning in the domains that comprise PTG before and after a traumatic event, to assess actual PTG. These assessments of functioning were direct measures of the domains believed to comprise PTG, including personal strength, appreciation of life, relationships with others, new possibilities, and spiritual growth. At posttrauma, participants completed the PTGI. The results revealed that PTGI scores were either weakly related or unrelated to changes in actual PTG. Further, PTGI scores were positively associated with coping efforts and increases in distress. A number of similar studies that have examined changes in actual PTG over time have consistently found that PTGI scores are unrelated changes in actual PTG (Boerner, Joseph, and Murphy, 2017; Kunz, Joseph, Geyh, and Peter, 2018; Ransom, Sheldon, and Jacobsen, 2008; Trevino, Naik, and Moye, 2016; Yanez, Stanton, Hoyt, Tennen, and Lechner, 2011, [20,21,22,23,24]; although, for an important exception, see Iimura and Taku, 2018, [25]). Hence, it appears the PTGI does not assess actual PTG, but rather *perceived* PTG. The distinction between actual PTG and perceived PTG is a crucial one, and it is paramount that researchers recognize this distinction (Jayawickreme and Blackie, 2014, [6]).

It is similarly crucial that we understand the nature of perceived PTG and how it differs from actual PTG. For instance, what is the value of perceived PTG? Is it something clinicians should try to foster? Social psychological research has found that overinflated perceptions of oneself are related to better psychological adjustment (Humberg et al., 2018, [15]). Alternatively, perhaps perceived PTG is something we should discourage—trauma research finds that perceived PTG is generally related to worse psychological adjustment (Boerner, Joseph, and Murphy, 2017; Helgeson et al., 2006; Shakespeare-Finch and Lurie-Beck, 2014, [3,19,20]). A second important question, from a more basic science approach, is whether perceived PTG is related to how much individuals perceive they have changed from pre- to posttrauma, regardless of how much of this change is attributable to the traumatic event. As mentioned earlier, distressed individuals tend to exaggerate how much they believe they have improved over time (McFarland and Alvaro, 2000, [17]). This tendency during times of distress makes it easy for individuals to conflate general perceived growth over time (see Brown et al., 2011; Wilson and Ross, 2001, [8,9]), with perceived growth resulting from trauma exposure (perceived PTG). A third question is whether perceived PTG reflects attempts of coping with the distress of the traumatic event. Convincing oneself that they have grown as a result of the trauma could help to alleviate anxiety and distress. Recall that Frazier et al. (2009), [7] found that PTGI scores were positively correlated with increases in distress and coping efforts. Numerous other studies have found that PTGI scores are positively related to increased coping efforts (Brooks, Graham-Kevan, Robinson, and Lowe, 2018; Michael and Cooper, 2013; Yeung, Lu, Wong, and Huynh, 2016, [26,27,28]) and reflect a neurotic and immature defensive style (Boerner et al., 2017, [20]). Hence, perceived PTG may be a coping mechanism—another cognitive strategy that is part of the earlier discussed ‘psychological immune system’ designed to maintain ego and well-being. 

## 2. The Present Study

In the current study, we used a research design very similar to the one employed by Frazier et al. (2009) [7]—a prospective design that involved assessments of actual PTG and distress levels in a large sample of undergraduates at the beginning (Time 1) and end (Time 2) of a semester. Similar to Frazier et al., we assessed actual PTG by administering six measures of functioning in the domains that comprise PTG, administered at Time 1 and Time 2. These six measures were the same measures used by Frazier et al. The first of these measures is the current standing version of the PTGI (C-PTGI; Frazier et al.), which mimics the items of the PTGI, but asks about their current level of functioning (as opposed to perceived changes as a result of a traumatic event). The C-PTGI is our primary measure of actual PTG, since it encompasses all of the PTG domains included on the PTGI (personal strength, relating to others, spirituality, appreciation of life, and new possibilities). The other five measures of actual PTG we used are measures that specifically assess one of the aforementioned PTG domains. Undergraduate samples report particularly high rates of trauma exposure (Anders, Frazier, and Shallcross, 2012, [29]), making this sample advantageous for a prospective study of trauma exposure. We tracked which participants experienced a traumatic event in the time in-between (‘intervening trauma’). Also similar to Frazier et al., we gave the PTGI at Time 2 for the intervening traumatic events, along with measures of coping and distress. A novelty we added to our research design is that at Time 2, we asked participants to report how they believe they were doing at Time 1. We could then compare these retrospective reports at Time 2 with how they actually responded to the same questions at Time 1. The difference between these two measures gives us a measure of perceived general change at Time 2. It is important to note that this is a measure of perceived *general* change over time, with no references to change resulting from trauma. This latter lack of reference to change resulting from a trauma is what distinguishes perceived general growth from perceived PTG. In summary, the current study design yields three primary measures of changes, (1) actual PTG, (2) perceived general growth, and (3) perceived PTG. We tested three hypotheses that follow from our aforementioned questions about the nature of perceived PTG. The first two hypotheses are based on the notion that perceived PTG is a different construct than actual or perceived change. The first hypothesis is that actual PTG and perceived general growth will be positively related to each other; by contrast, perceived PTG will be unrelated to these two measures. The second hypothesis is that actual PTG and perceived general growth will be positively related to changes in PTG-domain measures; by contrast, perceived PTG will be unrelated to changes in PTG-domain measures. The third hypothesis concerns the notion that perceived PTG is, at least partly, a coping mechanism. The third hypothesis is that actual PTG and perceived general growth will be related to decreases in distress and posttraumatic stress symptoms at Time 2, and unrelated to coping; by contrast, perceived PTG will be related to increases in distress and posttraumatic stress symptoms at Time 2, and increased use of coping.

## 3. Method

### 3.1. Participants

Participants (*N* = 282; 90 male) were recruited from a large university in the southern United States and received partial course credit for participating. The average age was 19.58 years (*SD* = 3.02, Range 18–40). The ethnicity profile of the final sample was 43% White, non-Hispanic, 22% African-American, 24% Hispanic, 6% Asian, 1% Native American, and 4% Other. A total of 221 (78%) of participants completed Time 2. There were no significant differences between completers (completed Time 1 and Time 2) and noncompleters (completed Time 1 only) on any of the variables included in the study, including age, gender, and ethnicity ( *p*s > 0.10 for all), with one exception. Completers reported significantly higher levels of distress on the depression, anxiety, and stress scales (DASS) (*m* = 34.26, *sd* = 11.84) than did noncompleters (*m* = 30.87, *sd* = 11.99), *t* (279) = 2.03, *p* = 0.04, Cohen’s *d* = 0.29. Sixty-four of the 221 participants (29%) who completed Time 2 reported an intervening trauma. This subsample was 76% female, with an average age of 19.68 years (*SD* = 3.49, Range 18–36), and was 44% White, non-Hispanic, 21% African-American, 22% Hispanic, 8% Asian, and 5% Other.

### 3.2. Measures

**Perceived PTG.** We used the PTGI (Tedeschi and Calhoun, 1996, [1]), which is the most commonly used measure of PTG. The PTGI consists of 21 items. The PTGI asks the respondent “the degree to which the change reflected in the question is true in your life as a result of your crisis”. Example items include “I developed new interests” and “I have more compassion for others”. In the current sample, α = 0.97.

**Actual PTG—Primary Measure**. We used the same measure of actual PTG that was used in Frazier et al. (2009), [7], the current standing version of the PTGI (C-PTGI; Frazier et al., 2009, [7]). The C-PTGI asks the same items as the PTGI but asks the respondent “the degree to which the statement is true in your life during the past two weeks”. Thus, the C-PTGI asks about general wellbeing in the same domains as the PTGI and makes no references to changes over time or any event. In the current study, the internal reliabilities were α = 0.89 at Time 1 and α = 0.95 at Time 2. Actual PTG was assessed by subtracting C-PTGI scores at Time 1 from C-PTGI scores at Time 2.

**Actual PTG—Secondary Measures**. We used the same five measures used by Frazier et al. (2009), [7]. For all measures, higher scores reflect more of that construct. The nine-item positive relations with others subscale of the psychological wellbeing scale (PWB; Ryff, 1989, [30]) was used to represent the PTGI domain of relating to others. The measure contains items such as “I have not experienced many warm and trusting relationships with others”. In the current study, the internal reliabilities were α = 0.82 at Time 1 and α = 0.83 at Time 2. The five-item presence of meaning subscale from the meaning in life questionnaire (MLQ; Steger, Frazier, Oishi, and Kaler, 2006, [31]) was chosen to reflect the PTGI new possibilities subscale (e.g., “My life has a clear sense of purpose”). In the current study, the internal reliabilities were α = 0.93 at Time 1 and α = 0.94 at Time 2. Two measures were selected to represent the PTGI appreciation of life domain—the five-item satisfaction with life scale (SWLS; Diener, Emmons, Larsen, and Griffin, 1985, [32]; e.g., “I am satisfied with my life”), and the six-item gratitude questionnaire-6 (GQ-6; McCullough, Emmons, and Tsang, 2002, [33]; e.g., “I have so much in life to be thankful for”). In the current study, the internal reliabilities were α = 0.89 at both times for the SWLS and α = 0.77 at Time 1 and α = 0.83 at Time 2 for the GQ-6. The domain of spiritual change was assessed using the 10-item religious commitment inventory (RCI-10; Worthington et al., 2003, [34]; e.g., “Religious beliefs influence all my dealings in life”). In the current study, the internal reliabilities were α = 0.96 at both times. For all measures, change scores from Time 1 to Time 2 were calculated to assess actual PTG.

**Perceived General Growth**. At Time 2 only, we gave the participants a modified version of the C-PTGI, which we call the C-PTGI-Past. The C-PTGI-Past is identical to the C-PTGI (including the items and the response scale), with the only difference occurring in the instructions. The C-PTGI asks respondents to refer to the past two weeks, whereas the C-PTGI-Past asks respondents to refer to “…your life at the beginning of the semester”. Hence, the C-PTGI-Past assesses respondents’ perception of how they think they were doing at Time 1. We then subtracted the C-PTGI-Past total score from the C-PTGI administered at Time 2 total score to assess perceived general change.

**Coping.** We used the brief COPE (BCOPE; Carver, 1997, [35]) to measure coping behaviors. There are 14 distinct subscales, but a common methodology is to use three subscales—problem-focused coping (e.g., “I’ve been taking action to try to make the situation better”), emotion-focused coping (e.g., “I’ve been getting emotional support from others”, and avoidant-focused coping (e.g., “I’ve been refusing to believe that it has happened”). We also examined the positive reframing subscale (one of the 14 original subscales) for purposes of making direct comparisons to results reported in Frazier et al. (2009), [7]. No specific event was mentioned in the instructions, so in the current study, this measure assesses general tendencies to use the listed coping strategies. In the current study, the internal reliabilities were α = 0.81, 0.77, 0.79, and 0.73 for the avoidant, problem, emotion, and positive reframing subscales, respectively.

**Intervening Trauma**. We used the LEC-5 (Weathers et al., 2013 [36]) to assess whether participants experienced a traumatic event between Time 1 and Time 2. Participants were asked to indicate if they experienced each of 17 potentially traumatic events since Time 1. If a participant indicated more than one event, they were asked to indicate which event was the most traumatic. Participants who selected one or more of the 17 potentially traumatic events were placed in the ‘intervening trauma’ group; participants who did not experience any of these events were asked to briefly indicate the most stressful event they did experience since Time 1 and were placed in the ‘no intervening trauma’ group.

**Posttraumatic Stress Symptoms (PTSS)**. We used the 20-item PTSD checklist (PCL-5; Weathers et al., 2013 [36]), a commonly used measure of PTSD symptoms, to assess distress specific to the intervening trauma. Example items include “Feeling jumpy or easily startled” and “Being ‘superalert’ or watchful or on guard”. In the current sample, α = 0.96.

**Distress**. We used the 19-item depression, anxiety, and stress scales (DASS; Lovibond and Lovibond, 1995, Tra [37]) to assess general distress. Although there are three subscales (depression, anxiety, and stress), we chose to use the total score. Example items include “I found it difficult to relax” and “I felt I was close to panic”. In the current sample, α = 0.93.

### 3.3. Procedure

We conducted a cross-sectional, longitudinal, and prospective research design in which participants completed measures during the first three weeks of the semester (Time 1) and the last three weeks of the semester (Time 2). At Time 1, participants completed the C-PTGI, the five secondary measures that assess the domains of PTG, the BCOPE (coping), and the DASS (distress). At Time 2, participants completed the DASS, followed by the LEC-5, which was used to identify each participant’s most stressful event they experienced since Time 1. A frequency count of intervening traumatic events is reported in Table 1. Participants then completed the PCL-5 (PTSS) in reference to the most stressful or traumatic event they identified on the LEC-5. Participants then completed the C-PTGI-Past, the C-PTGI, the PTGI (in reference to the most stressful or traumatic event they identified on the LEC-5), and the five secondary measures that assess the domains of PTG. Participants completed all measures on a computer in a laboratory. All subjects gave their informed consent for inclusion before they participated in the study. The study was conducted in accordance with the Declaration of Helsinki, and the protocol was approved by the Ethics Committee of the University of North Texas (15-347).

## 4. Results

Means and standard deviations for all variables are included in Table 2. We began with some preliminary analyses to confirm the validity of the C-PTGI. Frazier et al. 2009 [7], found the C-PTGI was correlated (minimum *r* = 0.38) with the five secondary measures of actual PTG, cross-sectionally at Time 2. We wanted to make certain we replicated these findings in the current sample, with some additional analyses. Since change scores in our measures are a central part of our hypotheses, we (1) examined correlations between the C-PTGI and the five secondary measures of actual PTG at Time 2 (i.e., attempted to replicate the findings of Frazier et al. [7] in the current sample), (2) examined the same cross-sectional correlations at Time 1, (3) examined correlations between change scores in these measures, and (4) included distress in the analyses to further test the validity of the C-PTGI. As can be seen in Table 3, the C-PTGI was significantly correlated with all five secondary measures and distress in the expected directions, at both Time 1 and Time 2. We examined these correlations in both the full sample and the intervening trauma sample. We included the analyses in the intervening trauma sample only because later, when we conduct analyses with PTGI scores, only the intervening trauma sample will be relevant. As can be seen in Table 4 (left column of Full Sample), for the full sample, changes in C-PTGI scores were significantly correlated with changes in four of the five secondary measures (religious commitment being the exception) and changes in distress, in the expected directions. For the intervening trauma sample only, we observed the exact same pattern of results. Hence, the C-PTGI demonstrated good validity in both cross-sectional and longitudinal analyses, and in the full sample and intervening trauma sample only. Given the use of the unique measure CPTGI-Past in this study, we had an opportunity to examine the accuracy of participants’ perceptions of their past well-being. To accomplish this task, we examined the correlation between the C-PTGI-Past at Time 2 (i.e., at Time 2, how participants thought they were doing at Time 1) and the C-PTGI at Time 1 (i.e., how they actually were doing at Time 1). The resulting correlation was significant, *r* (218) = 0.59, *p* < 0.001 (i.e., accounting for 35% of variance in each other). Hence, participants’ reflections of how they think they were doing three months earlier only accounted for 35% of the variance in how they were actually doing three months prior.

To test our hypotheses, we examined correlations between actual PTG, perceived general growth, perceived PTG, and changes in the secondary measures of actual PTG, changes in distress, and coping. Since the PTGI is intended to assess growth from traumatic events (and to make direct comparisons with the results of Frazier et al. (2009) [7], who only examined participants who reported an intervening trauma), we used the intervening trauma sample only to test these hypotheses. However, for the purposes of comparison, we included the results of the full sample and the intervening trauma sample in Table 3. The results supported the first hypothesis, regardless of which sample we used. As can be seen in Table 4, perceived general growth and actual PTG were significantly positively correlated with each other, suggesting that participants’ perceptions of change in the domains that comprise PTG were accurate. Further, perceived PTG was not significantly related to either actual PTG or perceived general growth.

The results mostly supported the second hypothesis. As can be seen in Table 4, in the intervening trauma sample only, actual PTG was significantly related to changes in four out of five secondary measures of actual PTG (the exception being changes in religious commitment). Further, perceived general growth was significantly related to changes in three out of five secondary measures of actual PTG (exceptions being changes in religious commitment and meaning in life). Lastly, perceived PTG was not significantly related to any of the changes in the five secondary measures of actual PTG. 

The results mostly supported the third hypothesis. As can be seen in Table 4, both actual PTG and perceived general growth were significantly negatively related to changes in distress (but not significantly related to PTSS at Time 2) and unrelated to all forms of coping. Further, perceived PTG was significantly positively related to changes in distress and PTSS at Time 2. We hypothesized that perceived PTG would be related to coping, but this was only true for avoidance coping. 

## 5. Discussion

The preliminary results demonstrated the solid validity of our primary measure of actual PTG, which replicated and extended similar findings reported in Frazier et al. (2009) [7]. Participants’ perceptions of how much they had generally changed across the semester were tied to growth resulting from a stressful event—perceived general change was strongly positively related to actual PTG. However, the divergence in associations involving perceived PTG was stark. Actual PTG and perceived general growth were positively related, and both were negatively related to changes in distress and unrelated to coping. However, perceived PTG in the early aftermath of a traumatic experience appeared to reflect more of a coping process than an accurate recall of change—perceived PTG was unrelated to changes in actual PTG, positively related to changes in distress, and positively related to avoidance coping. This pattern of results supports the notion that perceived PTG is a separate construct from actual change or even perceived general change.

Our results support the Janus–Face model of PTG (Maercker and Zoellner, 2004 [38]), which posits that PTG has two sides—a constructive side reflecting authentic growth, and an illusory side rooted in cognitive biases designed to cope with distress. Our data suggest PTGI scores and other similar self-reports of PTG assess, at least in part, the illusory side of PTG. This notion underscores the importance of making a clear distinction between perceived PTG and actual PTG. Much has been made about the difficulty of assessing how much one has changed as a result of a specific event (Coyne and Tennen, 2010 [10]), with one of the steps being able to accurately assess how you were doing prior to the event. In our preliminary analyses, we found that participants were only somewhat accurate (35% of variance explained) in assessing how they were doing three months earlier. Oftentimes, researchers ask participants to judge how much they have grown from events that occurred years ago. It is highly likely that participants’ accuracy level decreases precipitously as the time since the event similarly increases. It is difficult for participants to accurately judge how much of their growth can be attributed to a specific event, when their ability to judge their general growth is erroneous. We also assessed whether perceived PTG has value and should be encouraged. Unfortunately, our data, in concert with other data, finds that perceived PTG is related to increases in distress, which suggests perceived PTG should be discouraged. However, despite these findings, it is still possible that perceived PTG is beneficial. Consider a scenario in which an individual is exposed to a trauma, which results in significant increases in distress. As a coping mechanism, the individual then unknowingly convinces him/herself that s/he has grown and benefitted from the trauma in important ways (i.e., created perceived PTG). This form of coping is effective in alleviating some of the distress, but not all of it. In research studies, such individuals would exhibit positive relationships between distress, coping, and PTGI scores, just as we found. These correlations would be interpreted by the researcher as perceived PTG being unhealthy, when in reality, the creation of perceived PTG was an effective coping mechanism. If such a coping strategy were ineffective when the trauma-related distress levels are extreme, researchers would observe an inverted-U relationship between perceived PTG and PTSS. Indeed, a meta-analysis of perceived PTG and PTSS found this exact relationship (Shakespeare-Finch and Lurie-Beck, 2014 [3]). We need experimental designs likely involving interventions to further examine the value of perceived PTG.

The current study has some limitations that are worthy of note. The first limitation is the relatively short time period between the intervening trauma and the assessment of perceived PTG. Genuine PTG can take months or sometimes years after a trauma to occur. Prospective studies that have longer-term follow-ups are ideal to assess changes that result from trauma. The second limitation is our limited statistical power. The intervening trauma sample in the current study (*N* = 64) was only slightly more than half the sample size in the Frazier et al. study (*N* = 120). However, the full sample had a much larger sample size and we obtained the same pattern of results, regardless of which sample we examined. A third limitation is we did not replicate all of the findings reported in Frazier et al. (2009) [7]. First, Frazier et al. found a small but significant positive correlation between the Time 2 PTGI scores and change in C-PTGI scores (*r* = 0.22); this same correlation was not significant in the current study (*r* = −0.22, *ns*). This discrepancy suggests multiple possibilities, including that this association is spurious and/or this association varies, depending on one or more factors. Second, Frazier et al. found a positive correlation (*r* = 0.29) between Time 2 PTGI scores and changes in religious commitment (one of the five secondary measures of actual change); this same correlation was not significant in the current study (*r* = 0.08, *ns*). Third, Frazier et al. found Time 2 PTGI scores were strongly correlated with Time 2 positive reframing coping (*r* = 0.52); this correlation was not significant in the current study (*r* = 0.01, *ns*). One possibility for this failure to replicate is that the way we used the BCOPE was as an assessment of general coping tendencies, whereas Frazier et al. used the BCOPE to examine coping specific to the traumatic event. A fourth limitation is our use of an undergraduate sample. An undergraduate sample has the advantage of elevated rates of trauma exposure but comes at the cost of restricted ranges of age, intelligence, and socioeconomic status. The fifth limitation is that, according to Calhoun and Tedeschi (1998) [39], only events that are ‘seismic’ and severely shake an individual’s core beliefs are capable of producing PTG. We used events considered traumatic by the LEC-5, which is based on current diagnostic criteria, to comprise our subsample of participants who could potentially experience PTG. This use of traumatic events is only a proxy for the conceptual foundation expressed by Calhoun and Tedeschi. A sixth limitation is that we used different methodologies to measure perceptions of change for our two constructs of perceived general change and perceived PTG. For the former construct, we asked participants at Time 2 to rate how they were doing now and how they think they were doing at Time 1 and subtracted the latter from the former. For the latter construct, we used the PTGI, which is a one-time measure that asks participants to rate the degree of change as a result of the traumatic event.

The measurement of psychological constructs is a challenging endeavor. Unfortunately, we cannot slip a thermometer under a participant’s tongue and get an objective, precise measurement of PTG. In psychology, more times than not, we have to rely on self-reports for measurement. There is growing empirical evidence that self-reports of PTG are contaminated by a number of motivated cognitive biases. It is crucial that we recognize the distinction between perceived PTG and actual PTG. The results of the current study support this distinction. We encourage researchers to recognize this distinction and continue examining the nature and value of perceived PTG versus actual PTG. Such research is vital to our understanding of the nature of PTG and our ability to help trauma victims recover, grow, and flourish.

## Figures and Tables

**Table 1 behavsci-09-00010-t001:** Frequency of intervening traumatic events.

Traumatic Event	
Natural Disaster	8
Fire or Explosion	1
Transportation Accident	9
Serious Accident	2
Exposure to Toxic Substance	1
Physical Assault	5
Assault with a Weapon	0
Sexual Assault	1
Unwanted Sexual Experience	7
Combat	0
Captivity	1
Life Threating Illness or Injury	2
Severe Suffering	2
Violent Death	1
Accidental Death	3
Serious Harm You Caused	1
Other Traumatic Experience	20

**Table 2 behavsci-09-00010-t002:** Means and standard deviations of all variables.

	Full Sample		Intervening Trauma Only	
	Time 1	Time 2	Time 1	Time 2
C-PTGI	94.82(14.66)	92.84(21.08)	95.64(14.95)	90.97(19.04)
C-PTGI-Past	-	91.89(20.02)	-	94.40(18.08)
Perceived PTG (PTGI)	-	36.68(23.46)	-	37.67(20.16)
Satisfaction with Life	22.57(7.17)	22.68(7.51)	21.59(7.53)	21.00(7.30)
Gratitude	35.17(5.16)	35.27(6.12)	34.61(5.53)	34.28(6.96)
Positive Relations	38.40(7.76)	37.84(8.38)	38.36(8.23)	36.50(8.23)
Religious Commitment	21.69(11.78)	21.11(11.85)	21.21(11.47)	20.65(11.08)
Meaning in Life	25.88(7.17)	25.75(7.83)	26.33(6.51)	25.71(8.14)
Distress	34.18(11.84)	33.29(11.89)	35.19(11.72)	36.94(12.22)
PTSS (PCL-5)	-	18.03(18.54)	-	24.00(18.20)
Positive Reframing Coping	5.37(1.69)	-	5.82(1.57)	-
Problem Coping	19.90(5.31)	-	20.55(5.34)	-
Emotion Coping	24.26(5.73)	-	26.22(5.37)	-
Avoidance Coping	18.04(5.28)	-	18.80(5.53)	-

Note: *N* for full sample at Time 1 = 282; *N* at full sample Time 2 = 219-221; *n* for Trauma Only at Time 1 = 64; *n* for Trauma Only at Time 2 = 66. ** *p* < 0.01, *** *p* < 0.001. C-PTGI = current standing of posttraumatic growth. PTSS = posttraumatic stress symptoms.

**Table 3 behavsci-09-00010-t003:** Correlations between the C-PTGI, secondary measures of actual PTG, and distress at Time 1 and Time 2.

	Full Sample		Intervening Trauma Only	
	Time 1	Time 2	Time 1	Time 2
Satisfaction with Life	0.62 ***	0.59 ***	0.77 ***	0.53 ***
Gratitude	0.50 ***	0.56 ***	0.43 ***	0.46 ***
Positive Relations	0.51 ***	0.56 ***	0.63 ***	0.52 ***
Religious Commitment	0.43 ***	0.33 ***	0.53 ***	0.33 **
Meaning in Life	0.68 ***	0.58 ***	0.67 ***	0.57 ***
Distress	−0.39 ***	−0.44 ***	−0.51 ***	−0.48 ***

Note: *N* for full sample at Time 1 = 282; *N* at full sample Time 2 = 219–221; *n* for Trauma Only at Time 1 = 64; *n* for Trauma Only at Time 2 = 66. ** *p* < 0.01, *** *p* < 0.001. C-PTGI = current standing of posttraumatic growth.

**Table 4 behavsci-09-00010-t004:** Correlations between actual change, perceived general change, and perceived PTG scores and changes in PTG-domains, distress, and coping in the full sample and trauma sample only.

	Full Sample (N = 214–218)				Intervening Trauma Only (N = 61–64)			
	M(SD)	Actual PTG	Perceived General Growth	Perceived PTG	M(SD)	Actual PTG	Perceived General Growth	Perceived PTG
Actual PTG	−1.99(18.72)	-	0.57 ***	−0.12	−4.67(18.01)	-	0.67 ***	−0.22
Perceived General Growth	1.13(16.69)		-	−0.11	−2.90(15.71)		-	−0.24
Time 2 Perceived PTG	36.68(23.46)			-	37.67(20.16)			-
Change in Satisfaction with Life	0.10(4.93)	0.35 ***	0.32 ***	−0.10	−0.59(5.18)	0.46 ***	0.36 **	−0.12
Change in Gratitude	0.10(4.67)	0.26 ***	0.12	−0.07	−0.33(4.88)	0.29 *	−0.01	−0.14
Change in Positive Relations	−0.56(6.66)	0.38 ***	0.35 ***	−0.07	−1.86(5.95)	0.40 ***	0.32 **	−0.28
Change in Religious Commitment	−0.58(6.49)	0.12	0.02	−0.05	−0.56(5.53)	0.07	0.06	0.09
Change in Meaning in Life	−0.13(4.95)	0.37 ***	0.19 **	−0.07	−0.61(4.64)	0.36 **	0.11	−0.12
Change in Distress (DASS)	−0.89(9.91)	−0.32 ***	−0.39 ***	0.11	1.75(10.22)	−0.27 *	−0.35 **	0.43 **
Time 2 PTSS (PCL-5)	18.03(18.54)	−0.16 *	−0.14 *	0.30 ***	24.00(18.20)	−0.17	−0.16	0.38 **
Time 1 Positive Reframing Coping	5.37(1.69)	−0.01	−0.03	0.03	5.82(1.57)	−0.01	−0.13	0.02
Time 1 Problem Coping	19.90(5.31)	0.02	0.03	0.10	20.55(5.34)	−0.07	−0.09	0.11
Time 1 Emotion Coping	24.26(5.73)	−0.02	−0.02	0.15 *	26.22(5.37)	0.02	−0.11	0.22
Time 1 Avoidance Coping	18.04(5.28)	−0.05	0.11	0.25 *	18.80(5.53)	−0.05	0.10	0.37 **

Note: * *p* < 0.05, ** *p* < 0.01, *** *p* < 0.001. PTSS = posttraumatic stress symptoms. PTG = posttraumatic growth. Time 2 Perceived PTG and Time 2 PTSS were completed in reference to the most stressful event experienced between Time 1 and Time 2.

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
