# Peer review of "Perceptions of Change after a Trauma and Perceived Posttraumatic Growth: A Prospective Examination"

_behavsci, 2019, doi:10.3390/bs9010010_

Round 1

Reviewer 1 Report

This study addresses the important issue of the construct validity of self-reported PTG.  The attempt to replicate – and extend - a prior study was a strength. The manuscript would be improved by a clearer description of the constructs and by removing the analyses related to coping. Comments on each section of the paper are outlined below.

Abstract

1.       Line 30, I don’t believe it is accurate to refer to “increases in perceived PTG” because this variable is only assessed at T2.

Introduction

2.       In the second full paragraph, the authors might want to add research showing that actual change and perceived change also are not strongly related in other domains (e.g., personality).  Thus, lack of correspondence between the PTGI and change scores is a more general phenomenon.

3.       I am very familiar with the issues addressed in this research but I found the terminology a bit hard to follow. I encourage the authors to use slightly more descriptive terms than actual growth, perceived growth, and perceived PTG. It is also awkward to describe changes in PTG domains as “changes in actual growth” (see lines 81, 85). Referring to changes in PTG-related domains seems more straightforward. I had to read the section describing the hypotheses on pp. 6-7 several times in an effort to keep these terms straight. In this section it would also be helpful for the authors to clarify that there are two measures of “actual growth” – change in C-PTGI scores and change in measures of PTG-related domains. The authors refer to the former but not the latter as “actual growth” but they both seem to measure “actual growth” or at least changes in measures from pre to postevent. It would also be helpful to clarify why the C-PTGI is conceptualized as the “primary” measure.

4.       Line 101: The authors might also want cite a more recent meta-analysis showing positive relations between PTG and PTSD (Shakespeare & Finch, 2014).

5.       Lines 108-109: One of the authors’ questions is whether “perceived PTG reflects attempts of coping with the distress of the traumatic event.” It is puzzling that they measured coping at T1 prior to the intervening traumatic event. This was also inconsistent with the Frazier et al. study they are trying to replicate. This is a major flaw with regard to this specific research question. On line 126, they state that coping was assessed at T2 and this is also implied in the description of the hypotheses related to coping on lines 142-146. However, coping is not included as a T2 measure on p. 10.

6.       In the description of the third hypothesis, it would be helpful to clarify the direction of the expected changes (e.g., actual growth and perceived growth will be related to decreases in distress and posttraumatic stress symptoms at Time 2 vs. “actual growth and perceived growth will be negatively related to changes in distress and posttraumatic stress symptoms at Time 2.”

Method

7.       Line 158, add effect size for difference between completers and noncompleters.

8.       On pp. 7-8, the authors refer to primary and secondary measures of actual growth. As mentioned, this should be clarified and explained earlier.

9.       On p. 9, in the description of the coping measure the referent is not clear (i.e., what “situation” are they referring to when they are describing their coping?). If it is not the intervening event then I don’t believe this measure addresses the research question and I don’t think these analyses should be included.

10.   On line 219, the authors state that the PTSD questions were answered with regard to the “intervening trauma” but, based on Table 2, it appears they were answered with regard to both traumas and other events.

Results

11.   It would be helpful to add some descriptive information on the types of intervening traumas (and other events) reported by the sample.

12.   Table 1: I am curious how the sample size for the intervening trauma group was bigger at T2 than at T1.

13.   Lines 274-5: It is curious that the relation between change in C-PTGI scores (what the authors are referring to as “actual growth”) has the same correlation with the PTGI as in the Frazier et al study but in the opposite direction (-.22 vs. .22). Although neither correlation is significant they are likely significantly different from each other. This is mentioned in the discussion but deserves further elaboration. Also I believe the correlation in this study is reported inaccurately on line 339.

Discussion

14.   Lines 297-8: The finding that students’ reports of change were related to actual change could use some more commentary including the extent to which it is consistent with prior research.

15.   Line 308: What aspects of the findings indicate the constructive side of perceived PTG reflecting actual growth? I did not understand the explanation the authors offered on lines 315-325.

16.   The limitation regarding the small sample size was mentioned twice on pp. 15-16. The limitation should be framed in terms of power not in relation to the sample size of the Frazier et al study.

Author Response

Abstract

1.       Line 30, I don’t believe it is accurate to refer to “increases in perceived PTG” because this variable is only assessed at T2.

Good catch. Right before I read the reviews, I re-read the paper (which I tend to do) and this mistake actually jumped out at me too. It now says ‘….higher levels of PTG”. We made the same mistake when mentioning coping in the next sentence, which we also fixed. 

Introduction

2.       In the second full paragraph, the authors might want to add research showing that actual change and perceived change also are not strongly related in other domains (e.g., personality).  Thus, lack of correspondence between the PTGI and change scores is a more general phenomenon.

Excellent point. In the aforementioned paragraph, we added, “Additionally, empirical evidence finds that people’s perception of how their personality has changed over time is, at best, only modestly related with any actual personality change (Costa & McCrae, 1989; Woodruff, 1983) and is often skewed towards beliefs of positive, self-enhancing changes (Fleeson & Heckhausen, 1997).” We agree that this is an important point to add.

3.       I am very familiar with the issues addressed in this research but I found the terminology a bit hard to follow. I encourage the authors to use slightly more descriptive terms than actual growth, perceived growth, and perceived PTG.

We agree that the terminology is hard to follow. We changed all instances of ‘actual growth’ to ‘actual PTG’ to make it more specific. We also changed all instances of ‘perceived change’ to ‘perceived general change’ to similarly make this term more specific. Hence the three terms we now use are actual PTG, perceived PTG, and perceived general change. We believe this makes following these terms much easier.

It is also awkward to describe changes in PTG domains as “changes in actual growth” (see lines 81, 85). Referring to changes in PTG-related domains seems more straightforward. I had to read the section describing the hypotheses on pp. 6-7 several times in an effort to keep these terms straight.

We can see why someone would say this and we agree it is reasonable. However, our paper is closely tied to the Frazier et al (2009) paper. In that paper, they specifically use the term ‘actual growth’. We want to use the same term to be consistent. As stated previously, we now use the term ‘actual PTG’ to be more specific, and we feel this term is similar enough to the term used in Frazier et al.

In this section it would also be helpful for the authors to clarify that there are two measures of “actual growth” – change in C-PTGI scores and change in measures of PTG-related domains. The authors refer to the former but not the latter as “actual growth” but they both seem to measure “actual growth” or at least changes in measures from pre to postevent. It would also be helpful to clarify why the C-PTGI is conceptualized as the “primary” measure.

Good point. Done.

4.       Line 101: The authors might also want cite a more recent meta-analysis showing positive relations between PTG and PTSD (Shakespeare & Finch, 2014).

Done.

5.       Lines 108-109: One of the authors’ questions is whether “perceived PTG reflects attempts of coping with the distress of the traumatic event.” It is puzzling that they measured coping at T1 prior to the intervening traumatic event. This was also inconsistent with the Frazier et al. study they are trying to replicate. This is a major flaw with regard to this specific research question. On line 126, they state that coping was assessed at T2 and this is also implied in the description of the hypotheses related to coping on lines 142-146. However, coping is not included as a T2 measure on p. 10.

This is a very legitimate point. We measured coping at Time 1, but not Time 2, which agree is a flaw. Ideally, we would have assessed coping at Time 2, as Frazier et al did. In the middle of the paragraph “Present Study” where we first state that we use a measure of coping, we added, “An important note is that Frazier et al. assessed coping at Time 2, whereas we only assessed coping at Time 1 and did not reference any specific stressful event. Hence our measure of coping assesses general tendencies to use certain coping strategies, whereas in the Frazier et al. study, coping was assessed specific to the traumatic event under investigation.” We address this issue further in your point #9 below.

6.       In the description of the third hypothesis, it would be helpful to clarify the direction of the expected changes (e.g., actual growth and perceived growth will be related to decreases in distress and posttraumatic stress symptoms at Time 2 vs. “actual growth and perceived growth will be negatively related to changes in distress and posttraumatic stress symptoms at Time 2.”

Done.

Method

7.       Line 158, add effect size for difference between completers and noncompleters.

Done.

8.       On pp. 7-8, the authors refer to primary and secondary measures of actual growth. As mentioned, this should be clarified and explained earlier.

As mentioned earlier, this is done.

9.       On p. 9, in the description of the coping measure the referent is not clear (i.e., what “situation” are they referring to when they are describing their coping?). If it is not the intervening event then I don’t believe this measure addresses the research question and I don’t think these analyses should be included.

We thought about this issue quite a bit. You make a very good point. We definitely considered removing coping from the paper entirely. However, in the end, we decided the paper is stronger with it included, and we hope you will agree. As mentioned earlier, we now explicitly state in the paper that the coping measure does not mention any event, hence it assesses general tendencies to use different coping strategies. We believe our findings with how we assessed coping, combined with the results from Frazier et al., add to our understanding of perceived PTG and coping. We also added this point in the second-to-last paragraph of the Discussion in which we acknowledge the study limitations.

10.   On line 219, the authors state that the PTSD questions were answered with regard to the “intervening trauma” but, based on Table 2, it appears they were answered with regard to both traumas and other events.

You are correct that the PCL-5 was given specific to the intervening trauma, as we state on line 219. We are not certain what you are seeing in Table 2 that suggests the PCL-5 was answered to both traumas or other events. To make it crystal clear, in the note for Table 2, we added, “Perceived PTG and Time 2 PTSS were completed in reference to the most stressful event experienced between Time 1 and Time 2.”

Results

11.   It would be helpful to add some descriptive information on the types of intervening traumas (and other events) reported by the sample.

Good point. We added a new Table (Table 1) that reports the frequencies of the intervening traumatic events.

12.   Table 1: I am curious how the sample size for the intervening trauma group was bigger at T2 than at T1.

We thought that might raise an eyebrow. It is due to missing data. Two participants in this group had some missing data at Time 1.

13.   Lines 274-5: It is curious that the relation between change in C-PTGI scores (what the authors are referring to as “actual growth”) has the same correlation with the PTGI as in the Frazier et al study but in the opposite direction (-.22 vs. .22). Although neither correlation is significant they are likely significantly different from each other. This is mentioned in the discussion but deserves further elaboration.

We don’t have a good explanation for the discrepancy. We offer two possibilities in the Discussion.

Also I believe the correlation in this study is reported inaccurately on line 339.

You are exactly right. We corrected this mistake.

Discussion

14.   Lines 297-8: The finding that students’ reports of change were related to actual change could use some more commentary including the extent to which it is consistent with prior research.

After further consideration, we believe it is misleading to say that perceptions of change were accurate based on our findings. Instead, we now more accurately report that perceived general change was strongly related to actual PTG in this section of the Discussion.

15.   Line 308: What aspects of the findings indicate the constructive side of perceived PTG reflecting actual growth? I did not understand the explanation the authors offered on lines 315-325.

We simply offered this rationale as a possible explanation of the findings. An analogy would be the relationship between aspirin and headaches. Researchers would likely find a positive relationship between aspirin and headaches (because only people with headaches take aspirin). However, what is really happening is that taking aspirin is effective at reducing headaches. The same phenomenon could be operating in our results. We found a positive relationship between perceived PTG and distress (perhaps because only people who are distressed construct illusory PTG), but what is really happening is the construction of perceived PTG is effective at reducing distress.

16.   The limitation regarding the small sample size was mentioned twice on pp. 15-16. The limitation should be framed in terms of power not in relation to the sample size of the Frazier et al study.

Done.

Reviewer 2 Report

Reviewed manuscript aims to clarify the distinction between actual/real and illusory/perceived posttraumatic growth (PTG), which is highly relevant topic in current state of the art. The veracity of PTG assessed by retrospective self-report measures is important research question with significant overlap into clinical care. The study is based on the data collected at two time points (beginning/end of the semester) with the sample of 282 undergraduate students who completed several measures of psychosocial adaptation, coping and checklist assessing the experience of traumatic situation during the semester. The study presents innovative methodology of assessing the veracity of PTG by a combination of PTGI, C-PTGI and C-PTGI-past and two different time points (before and after the traumatic event).

Comments:

-        Whole sample consists of 282 undergraduate students with about two thirds being females: is this disproportion given by their field of study with higher representation of women in general? Section “Participants” contains relevant characteristics of the whole sample, but only 64 of participants experienced trauma during semester (as assessed by LEC-5). This subgroup is analysed separately but with no details about its characteristics: proportion of males/females, age range etc.

-        Furthermore, I would appreciate more details about traumatic experiences of those 64 participants, because specific characteristics of the traumatic event (eg. temporal delimitation) can influence the process of adaptation.

-        Although the description of individual methods contains all relevant information about the measures and timing of assessment, it is quite hard for the reader to follow or keep in mind the subtle differences between the three types of variables concerning positive changes: perceived PTG, actual growth and perceived growth. Perhaps a table summarizing the timing of assessments and methods in the background of these variables could enhance the comprehensibility of the text.

-        Given the importance of subjective appraisal of traumatic situation described in posttraumatic growth literature, which seems to be more closely tied to PTG than objective evaluation, I would suggest the discussion of the possibility that not all of the 64 respondents indicating the presence of 1 or more traumatic situations on LEC-5 had to be really traumatized. Was this option taken into account during the scoring of LEC-5?

-        Although authors admit that one semester may be too short for trauma processing and genuine PTG may take longer time to occur, they suggest “perceived PTG may be more of a coping process than an accurate recall of posttraumatic change”. This conclusion could be articulated more specifically in order to take into account this time frame (e.g. perceived PTG in the early aftermath of trauma…). Helgeson, Reynolds & Tomich (2006; in the Literature list of the manuscript) suggested PTG assessed shortly after trauma may reflect cognitive strategy used to reduce stress, whereas PTG assessed after longer time lapse may reflect actual positive change.

Author Response

-        Whole sample consists of 282 undergraduate students with about two thirds being females: is this disproportion given by their field of study with higher representation of women in general?

This proportion (2/3 female) is very typical of Psychology Subject Pool samples.

Section “Participants” contains relevant characteristics of the whole sample, but only 64 of participants experienced trauma during semester (as assessed by LEC-5). This subgroup is analysed separately but with no details about its characteristics: proportion of males/females, age range etc.

Demographics on the subsample has been added to the end of the Participants section.

-        Furthermore, I would appreciate more details about traumatic experiences of those 64 participants, because specific characteristics of the traumatic event (eg. temporal delimitation) can influence the process of adaptation.

We added a new Table (Table 1) that reports the frequencies of the intervening traumatic events. We did not collect data on time since event for these events since the time window was very small (all events occurred within last 3 months).

-        Although the description of individual methods contains all relevant information about the measures and timing of assessment, it is quite hard for the reader to follow or keep in mind the subtle differences between the three types of variables concerning positive changes: perceived PTG, actual growth and perceived growth. Perhaps a table summarizing the timing of assessments and methods in the background of these variables could enhance the comprehensibility of the text.

We agree and Reviewer 1 made the exact same point. We offer you the same response we gave to Reviewer 1: We agree that the terminology is hard to follow. We changed all instances of ‘actual growth’ to ‘actual PTG’ to make it more specific. We also changed all instances of ‘perceived change’ to ‘perceived general change’ to similarly make this term more specific. Hence the three terms we now use are actual PTG, perceived PTG, and perceived general change. We believe this makes following these terms much easier.

-        Given the importance of subjective appraisal of traumatic situation described in posttraumatic growth literature, which seems to be more closely tied to PTG than objective evaluation, I would suggest the discussion of the possibility that not all of the 64 respondents indicating the presence of 1 or more traumatic situations on LEC-5 had to be really traumatized.

Agreed. At the end of the second-to-last paragraph of the Discussion, we added, “The fifth limitation is that, according to Calhoun and Tedeschi (1998), only events that are ‘seismic’ and severely shake an individual’s core beliefs are capable of producing PTG. We used events considered traumatic by the LEC-5, which is based on current diagnostic criteria, to comprise our subsample of traumatic events only. This measurement is only a proxy for the conceptual foundation expressed by Calhoun and Tedeschi.”

Was this option taken into account during the scoring of LEC-5?

There is no way to ‘score’ the LEC-5. It is simply a trauma history measure that can be used to assess trauma exposure.

-        Although authors admit that one semester may be too short for trauma processing and genuine PTG may take longer time to occur, they suggest “perceived PTG may be more of a coping process than an accurate recall of posttraumatic change”. This conclusion could be articulated more specifically in order to take into account this time frame (e.g. perceived PTG in the early aftermath of trauma…). Helgeson, Reynolds & Tomich (2006; in the Literature list of the manuscript) suggested PTG assessed shortly after trauma may reflect cognitive strategy used to reduce stress, whereas PTG assessed after longer time lapse may reflect actual positive change.

Agreed. We used your exact wording.

Reviewer 3 Report

Thank you for the opportunity to review the manuscript entitled “Perceptions of Change after a Trauma and Perceived Posttraumatic Growth: A Prospective Examination.” (Manuscript ID: behavsci-406067)

As a PTG researcher, I agree with the authors’ opinion that researchers should recognize the differences between actual PTG and perceived PTG. I have two important doubts regarding the manuscript. Please see my comments below.

1. Page 9, line 198. To assess perceived growth, the authors calculated a change score by subtracting the C-PTGI-Past total score at Time 2 from the C-PTGI total score at Time 2. I was not sure why this change score reflects perceived growth. Please explain this in a little more detail.

2. In the Results section, the authors should provide means and standard deviations for the change scores as in Frazier et al. (2009). Readers need to be able to evaluate the extent to which the participants changed on average, and its variance.

Author Response

1. Page 9, line 198. To assess perceived growth, the authors calculated a change score by subtracting the C-PTGI-Past total score at Time 2 from the C-PTGI total score at Time 2. I was not sure why this change score reflects perceived growth. Please explain this in a little more detail.

 As requested by Reviewers 1 and 2, we changed the terminology to make it clearer what the terms refer to in the study. We now use the term ‘perceived general growth’, which is a more accurate term to describe what results from subtracting the C-PTGI-Past total score at Time 2 from the C-PTGI total score at Time 2.

2. In the Results section, the authors should provide means and standard deviations for the change scores as in Frazier et al. (2009). Readers need to be able to evaluate the extent to which the participants changed on average, and its variance.

Good point. We added this data to Table 2.